# An EPAC1/PDE1C-Signaling Axis Regulates Formation of Leading-Edge Protrusion in Polarized Human Arterial Vascular Smooth Muscle Cells

**DOI:** 10.3390/cells8121473

**Published:** 2019-11-20

**Authors:** Paulina Brzezinska, Donald H. Maurice

**Affiliations:** Department of Biomedical and Molecular Sciences, Queen’s University, Kingston, ON K7L 3N6, Canada; paulina.brzezinska@queensu.ca

**Keywords:** exchange protein activated by cAMP, EPAC1, cyclic nucleotide phosphodiesterase, PDE1C, human arterial smooth muscle cells, HASMCs, migration, leading edge protrusion, cAMP, chemotaxis

## Abstract

Pharmacological activation of protein kinase A (PKA) reduces migration of arterial smooth muscle cells (ASMCs), including those isolated from human arteries (HASMCs). However, when individual migration-associated cellular events, including the polarization of cells in the direction of movement or rearrangements of the actin cytoskeleton, are studied in isolation, these individual events can be either promoted or inhibited in response to PKA activation. While pharmacological inhibition or deficiency of exchange protein activated by cAMP-1 (EPAC1) reduces the overall migration of ASMCs, the impact of EPAC1 inhibition or deficiency, or of its activation, on individual migration-related events has not been investigated. Herein, we report that EPAC1 facilitates the formation of leading-edge protrusions (LEPs) in HASMCs, a critical early event in the cell polarization that underpins their migration. Thus, RNAi-mediated silencing, or the selective pharmacological inhibition, of EPAC1 decreased the formation of LEPs by these cells. Furthermore, we show that the ability of EPAC1 to promote LEP formation by migrating HASMCs is regulated by a phosphodiesterase 1C (PDE1C)-regulated “pool” of intracellular HASMC cAMP but not by those regulated by the more abundant PDE3 or PDE4 activities. Overall, our data are consistent with a role for EPAC1 in regulating the formation of LEPs by polarized HASMCs and show that PDE1C-mediated cAMP hydrolysis controls this localized event.

## 1. Introduction

Agents that increase cyclic AMP (cAMP) signaling largely inhibit migration of arterial smooth muscle cells (ASMCs), including those ASMCs isolated from human arteries (HASMCs). For instance, studies have shown that the pan-cellular increases in cAMP caused by agents that activate all transmembrane adenylyl cyclases, including agents such as forskolin, or which inhibit all cellular cAMP-hydrolyzing phosphodiesterases (PDEs), like isobutyl-methyl-xanthine (IBMX), consistently reduce ASMC migration [1]. For these reasons, cAMP-elevating agents have long been seen as attractive agents through which to reduce ASMC migration in several conditions, including in-stent restenosis, where PDE4 inhibition reduces neointima formation and inhibits vascular cell adhesion molecule 1 (VCAM-1) expression and histone methylation in an exchange protein activated by cAMP (EPAC)-dependent manner [2,3] 

Interestingly, notwithstanding the observation that increased cAMP signaling results in reduced levels of ASMC migration, when the numerous steps involved in coordinating cellular migration are studied individually, it is found that they are equally likely to be inhibited or promoted [1,4,5,6]. For instance, while increased protein kinase A (PKA) activity reduces overall cellular migration, PKA activation promotes, rather than inhibits, the formation of cellular protrusions at the cellular front, a critical early step in polarizing cells for migration [5,6,7,8,9]. Although these dichotomous effects have not been studied systematically, it is likely that they arise due to the complexities involved in coordinating the recognized spatial and temporal selectivity of cellular cAMP signaling. Indeed, spatial and temporal compartmentation of cAMP signaling is made possible by the subcellular localization of the enzymes that synthesize cAMP (adenylyl cyclases), the enzymes that hydrolyze and inactivate cAMP (cyclic nucleotide phosphodiesterases; PDEs), and the dominant cAMP effectors (PKA and EPAC) [3,4,10,11,12,13]. Other factors that likely contribute to the dichotomous effects of cAMP on cell migration involve the interdependence of the individual steps (i.e., cell adhesion, protrusion, and retraction) required for cells to move efficiently [4,13,14]. Indeed, cell migration is an oscillatory process that requires a series of coordinated steps, in which extension or cell protrusion of the leading-edge initiates and maintains directional migration. Recently, multiple extracellular factors have also been shown to impact cell migration efficiency including such factors as the topography and deformability of the extracellular matrix and its composition [8,15]. Of relevance, and potentially directly linking cAMP signaling to these extrinsic factors, both PKA and EPAC1 are known to accumulate at the leading edge of migrating cells, including ASMCs, endothelial cells, and fibroblasts, and this compartmentation critically controls leading edge dynamics in an extracellular matrix protein-dependent fashion [5,6,8,9]. 

Since EPAC1 localizes at the front of migrating cells, and PKA activation within this domain is known to promote the formation of their leading-edge protrusions (LEP), we sought to investigate how EPAC1 impacted the formation of these dynamic structures. Overall, our findings are consistent with the idea that EPAC1 plays a critical role in promoting LEP formation in these HASMCs. Thus, RNAi-mediated silencing of EPAC1, or its pharmacological inhibition, decreased the formation of LEPs in HASMCs. In addition, we found that this effect was coordinated by calcium bound calmodulin (Ca^2+^/CaM)-regulated PDE inhibitors and silencing of PDE1C, but not by targeting the more dominant PDE3 or PDE4 activities. 

## 2. Materials and Methods

### 2.1. Cell Culture and siRNA Transient Transfections

Human arterial smooth muscle cells (HASMCs) were isolated from discarded unused portions of the internal thoracic artery in coronary artery bypass graft surgeries as described previously [16], from donor patients of Kingston General Hospital, as well as this, HASMCs were purchased from Cell Applications. For tissues obtained from Kingston General Hospital (KGH), their use in this research study (SURG-334-15; “Endothelial cell function in human hearts”) was approved by the Queen’s University Health Sciences and Affiliated Teaching Hospitals Research Ethics Board (HSREB). HASMCs were cultured in smooth muscle basal medium (SMBM) and smooth muscle growth medium bullet kit (SMGM-2) (Lonza), supplemented with 10% fetal bovine serum (FBS), cultured at 37 °C in 5% CO_2_, and used between passages 4–9. For siRNA transfection, HASMCs were cultured in basal SMBM containing Lipofectamine 3000 (Invitrogen) and siRNA (Sigma) in a 1:1 ratio, and media was changed 5 h post transfection with SMGM-2. Experiments were conducted 48 h post transfection. The following sequences of siRNAs were used, listed in Table 1. All siRNAs were purchased from Invitrogen. 

### 2.2. Chemotactic Leading Edge Protrusion (LEP) Assay

HASMCs resuspended in SMBM basal media were plated on the upper surface of gelatin-coated (ddH_2_0 supplemented with 0.25% gelatin (Biorad)), 3-μm, 24 mm^2^-diameter BD Falcon Corning^®^ FluoroBlok^TM^ cell culture inserts forming a monolayer, as described previously [9,17]. Chemotaxis was initiated by adding 0.5% FBS in SMBM media to the underside of the inserts to allow cells to form leading edge protrusions (LEPs) for 4 h. Pharmacological activators or inhibitors were added to the top of the insert prior to the addition of FBS to the underside of the inserts. The following drugs were used: CE3F4 (ToCRIS), 8-CPT-2′-*O*-Me-cAMP (Biolog), Compound 33 ((C33) a generous gift from Dr. Guy Breitenbucher; Dart Neurosciences), PF-04827736 (Sigma), Cilostamide (Calbiochem), and Ro 20-1724 (Calbiochem). To visualize the extent of LEPs, inserts were fixed with paraformaldehyde (4% (*v/v*)), rinsed with Hank’s Balanced Salt Solution (HBSS), and incubated for 1 h with phalloidin-tetramethylrhodamine B isothiocyanate (1:1000; Sigma) and DAPI (1:1000; Thermofisher) (0.3% bovine serum albumin (BSA) diluted in HBSS). Inserts were mounted on glass slides and the extent of LEPs were measured by quantifying the total fluorescence of phalloidin-TRITC on the bottom of the insert, as a measure of the total density of LEPs formed. In each case, 5 images were taken per transwell and these covered all 4 quadrants as well as the center of the transwell. In experiments in which we controlled for the number of cells applied to the top of the transwell, this was measured by counting the number of nuclei on the top of the transwell in a similarly unbiased and representative sampling of this structure. Images were captured with a Zeiss Axiovert S100 microscope and imaged with Slidebook software. LEP quantification was conducted by processing the images using Image Pro software, where the threshold tools were used to segment the LEPs followed by counting the pixel density of the area occupied by the LEPs. 

### 2.3. Immunoblotting

HASMCs monolayers were lysed using triton-based lysis buffer in (mM): 1.0% Triton X-100, 100 sodium pyrophosphate, 10 sodium β-glycerophosphate, 5 benzamidine, 10 sodium orthovanadate, 50 Tris-HCl, 100 sodium chloride, 1 EDTA, 5 magnesium chloride, 0.5 calcium chloride, 10 phenylmethylsulfonyl fluoride (PMSF), and the following protease inhibitors in µg/mL: 1 pepstantin A, 1 E-64, 5 bestantin, 1 aprotinin, and 2 leupeptin. Lysates were homogenized (20 G needle), centrifuged at 10,000 RPM, resolved by SDS-PAGE gels, transferred to PVDF membranes, and immunoblotted for the proteins of interest. The following anti-sera were used: anti-PDE1C (1:500; Fabgenix), anti-EPAC1 (cell signaling), anti-β-actin (1:10,000; Sigma), and anti-β-tubulin (1:1000; Sigma).

### 2.4. RNA Isolation, Reverse Transcription, and qPCR

HASMC RNA was isolated using the Qiagen RNeasy (Qiagen) mini kit as per the manufacturer’s instructions, followed by measurement of RNA purity and concentration using a Nanodrop 1000 (Thermo Scientific). cDNA was synthesized using a Qiagen Omniscript RT, according to the manufacturer’s instructions. qPCR reactions were performed using PowerUP^TM^ SYBR^TM^ Green Master Mix (Thermo Fisher Scientific) with 2 ng cDNA template, and the following primers were used, listed in Table 2. Thermocycler conditions were the following, using the QuantStudio 5 Real-Time PCR System: PCR stage: Step 1 95 °C, 15 min; Step 2 60 °C, 1 min repeated 40X. Melt curve stage: Step 1 95 °C, 15 min; Step 2 60 °C, 1 min, and Step 3 Dissociation, 95 °C, 1 s.

### 2.5. Statistical Analysis

All data presented were analyzed using GraphPad Prism Software and used for statistical analysis. Data in this study was collected from at least three independent experiments unless otherwise stated and presented as means ± (SEM). Statistical analysis between two groups was compared using an unpaired, two-tailed Student’s *t*-test and multiple comparisons were assessed using a 1- or 2-way analysis of variance (ANOVA), followed by the appropriate post-hoc test as indicated in the figure captions. A *p* value < 0.05 was considered significant. 

## 3. Results

### 3.1. Pharmacological Inhibition, or RNAi-Mediated Silencing, of EPAC1 Reduces Formation of Leading-Edge Protrusions (LEPs) in HASMCs

Using a combination of approaches, we assessed the role of EPAC1, the sole EPAC expressed in HASMCs [18], in coordinating the ability of these cells to generate polarized LEPs in response to a chemotactic gradient. Thus, treating HASMCs with an EPAC1-silencing siRNA decreased EPAC1 expression (Figure 1A), and antagonized the ability of these cells to generate LEPs (Figure 1B,C). Similarly, inhibiting EPAC1 pharmacologically with a selective EPAC1 inhibitor, CE3F4 (20 µM) [19,20], also markedly reduced the ability of HASMCs to generate LEPs in response to an FBS gradient (Figure 1D). In contrast, but consistent with the idea that EPAC1 is effectively activated in migrating HASMCs, the addition of the EPAC1 activating cAMP analogue, 8-CPT-2′-*O*-Me-cAMP (100 µM) [19], did not significantly alter the number of LEPs formed by these cells in our experiments (Figure 1E).

### 3.2. Selective Pharmacological Inhibition of HASMC PDEs Differentially Impacts Their Capacity to Generate LEPs

While previous studies have shown that pharmacological inhibition of the dominant HASMC cAMP PDEs, namely PDE1, PDE3, or PDE4, like PKA activation, reduced their migratory capacity [1,21], we hypothesized that selective pharmacological inhibition of PDE1, PDE3, or PDE4 might differentially impact the ability of these cells to form LEPs. Interestingly, while selective inhibition of HASMC PDE3 activity with cilostamide (5 µM) [1,22] reduced LEP formation in HASMCs, PDE4 inhibition with Ro 20-1724 (10 µM) did not (Table 3). Unexpectedly, pharmacological inhibition of PDE1 activity (C33, 1 µM) [23,24] in these cells markedly promoted the formation of LEPs in our experiments (Table 3). Although HASMCs have been reported by us and others to express both PDE1A and PDE1C gene-encoded enzymes, since PDE1C preferentially hydrolyzes cAMP compared to PDE1A, we next investigated the possibility that PDE1 inhibitors acted by inhibiting cAMP hydrolysis by PDE1C. Consistent with this hypothesis, silencing PDE1C (Figure 2A) increased the formation of LEPs (Figure 2B,C) and obviating the LEP producing effects of the PDE1 inhibitor, C33 (Table 4). 

### 3.3. Silencing HASMC EPAC1 Obviates PDE1 Inhibition-Directed LEP Formation

Since inhibition or silencing of the cAMP effector EPAC1 reduced LEP formation and PDE1 inhibition, or PDE1C silencing promoted the formation of these structures, we next investigated whether EPAC1 promoted LEP formation through a PDE1C-sensitive mechanism. To investigate this idea, we determined whether silencing EPAC1 would antagonize PDE1 inhibitor-mediated formation of LEPs in these cells. Thus, while PDE1 inhibition with either C33 (1 µM) or with PF-04827736 (1 µM) [25] promoted LEP formation in control HASMCs, neither of these PDE1 inhibitors were able to rescue LEP formation in EPAC1-silenced cells (Figure 3A,B) or in cells in which EPAC1 was inhibited (Figure 3C). Also, while silencing PDE1C promoted LEP formation in control cells, this basal effect and the ability of EPAC1 inhibition to promote LEP formation were lost in PDE1C-silenced HASMCs (Figure 3D). To obviate that our results reflected an effect related to a reduced adhesion of HASMCs to the upper surface of the transwells upon PDE1 inhibition or EPAC1-silencing, or loss of cells during the treatments periods, we counted HASMC nuclei which were present on the upper surfaces of the transwells, in which we detected changes in LEP numbers. As shown in (Figure 3E–H), inhibition or silencing of either PDE1C or EPAC1 did not significantly impact the number of HASMCs on the upper surface of the FluoroBlok^TM^ transwell in our studies. 

## 4. Discussion

Herein we show that silencing or inhibiting HASMC EPAC1 decreased the ability of these cells to generate LEPs in response to a chemotactic gradient. In addition, we show that this EPAC1 dependence for the generation of these actin-based leading-edge structures is regulated selectively by a source of intracellular cAMP that is regulated by PDE1C activity, but not by PDE3 or PDE4 activities. These data add to our understanding of the known dichotomous actions of cAMP, and its effectors, PKA and EPAC1, in the control of HASMC migration-associated activities. Specifically, these data show that EPAC1, like PKA, positively influences the formation of HASMC LEPs in the presence of a gradient and identify a potentially important role for PDE1C in these effects. 

Previous work has shown that modulating EPAC1 activity could either increase or decrease ASMC migration. Indeed, these earlier studies showed that EPAC1 activation with 8-CPT-2′-*O*-Me-cAMP promoted rat aortic SMC migration and facilitated ASMC accumulation into neointimal lesions formed following damage to murine femoral arteries [26]. Consistent with this, mice deficient in EPAC1 had reduced neointimal hyperplasia and ASMC migration under similar experimental conditions [26,27]. Interestingly, when SMCs isolated from human saphenous vein samples were used in experiments, EPAC1 activation negatively regulated their PDGF-induced migratory responses [28]. In this context, our work has begun to “unpack” these more global effects of EPAC1 in ASMCs and show that inhibiting EPAC1 activity, or markedly reducing its expression, inhibited LEP formation by migrating HASMCs. Our findings under conditions in which we pharmacologically activated EPAC1 were consistent with the idea that subjecting HASMC to a chemotactic gradient maximally activated EPAC1. Of course, further studies will be required to determine if this is the case, and more importantly, whether subjecting these cells to an FBS gradient selectively activates the fraction of EPAC1 localized at the leading edge of these cells. In addition, methodological differences between our study and others may also account for the differences which we observed. For instance, it was recently reported that EPAC1 could regulate SMC migration in a time- and concentration-dependent manner. Thus, it was reported that while high concentrations of the EPAC1 activator 8-CPT-2′-*O*-Me-cAMP (30–50 μM) significantly increased SMC migration compared to control untreated SMCs for up to 6 h, these effects were lost and replaced with inhibitory effects at longer time points [29]. 

With regards to the mechanism by which PDE1C regulates LEP formation in HASMCs, our data suggest that PDE1C regulates a distinct pool of cAMP than those regulated by PDE3 or PDE4. While some of our earlier and ongoing studies have shown that PDE1C is likely important in regulating the ability of PKA to impact LEP formation via effects on the store-operated calcium entry (SOCE) system in HASMCs [30], significant further work will be required in order to determine how PDE1C regulates EPAC1-mediated effects in these cells. In this context, previous work by others showed that the impact of SOCE in mediating cellular migration is influenced by the relative adhesive strength properties of the matrix and the cells [31]. Thus, it may be that PDE1C and EPAC1 will interact to regulate LEP formation differently when different matrix proteins are used. Indeed, EPAC1 may facilitate SMC LEP formation by interacting with different integrins based on the ECM protein tested [32]. In addition, it is likely that EPAC1 may impact SMC LEP formation by interacting with agents that control microtubule stability, since EPAC1 has been shown to regulate microtubule elongation [33]. For example, the low molecular weight GTPase Rap1, an EPAC1 effector, was shown to be activated at the leading edge of migrating vascular endothelial cells, and this was shown to accompany microtubule extension [34]. Furthermore, the importance of EPAC1 in regulating cell migration and microtubule stability was also previously reported, when inhibition of EPAC1 disrupted microtubule organization [35]. Another recent study identified the importance of graded cAMP signaling in mediating axonal guidance by promoting microtubule growth and membrane protrusion [36]. Therefore, future studies will be required to determine the impact of EPAC1 on microtubule stability in mediating ASMC protrusion and migration, and perhaps in other systems as well. Given the importance of Ca^2+^ signaling in guiding directed cell migration in processes, such as mesodermal sheet migration and gastrulation, axonal growth cone steering in developing neurons, and metastasis [36,37,38], the PDE1C/EPAC1 axis supports a connection by which Ca^2+^ and cAMP signaling systems may interact with one another locally to guide these physiological and pathological processes. In the context of HASMC LEP formation, PDE1C is known to be induced in migratory and proliferative vascular SMCs [21,39,40], thus this signaling axis provides a potential molecular target to mitigate vascular diseases where SMC migration is dysregulated, such as atherosclerosis and restenosis. 

## Figures and Tables

**Figure 1 cells-08-01473-f001:**
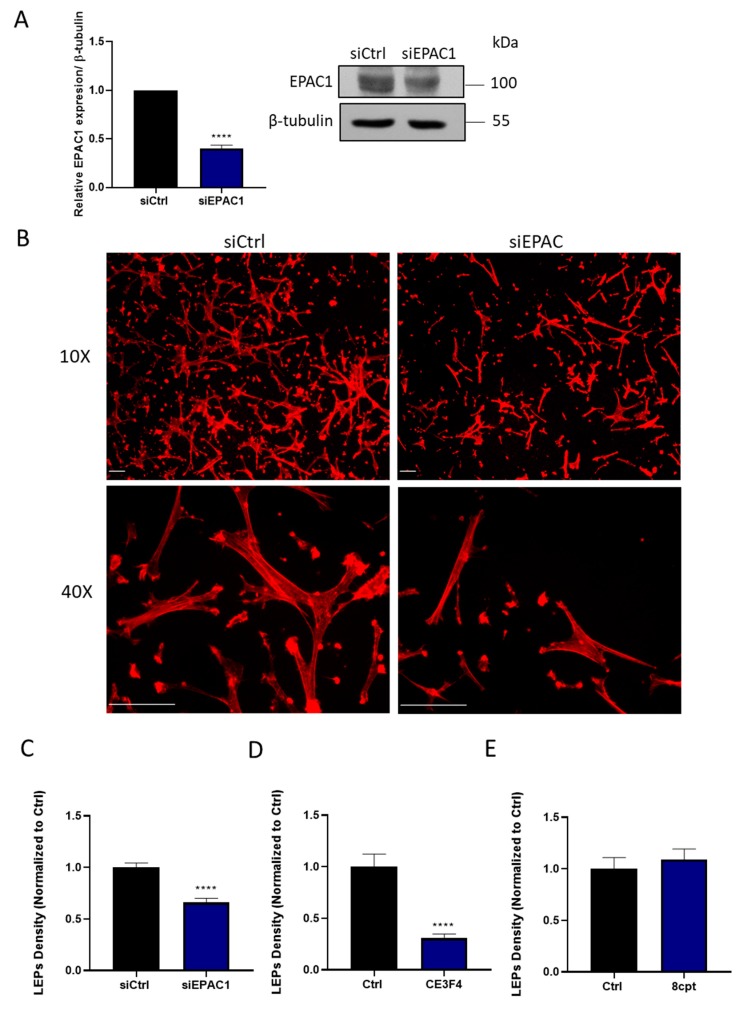
EPAC1 silencing or inhibition decreases leading-edge protrusion (LEP) formation by HASMCs. (**A**) Detection of EPAC1 by immunoblotting of samples obtained from a representative experiment in which HASMC were treated with siCtrl or siEPAC1 for 48 h is shown. Quantitating levels of EPAC1 in similar samples obtained from n = 3 independent experiments showed that siEPAC1 transfection significantly reduced EPAC1 levels when normalized to tubulin, as assessed using the Student’s unpaired *t*-test, **** *p* < 0.0001. **(B)** Representative images, obtained at either 10× or 40× magnification, of actin-stained LEPs detected on the lower levels (bottom) of FluoroBlok^TM^ transwells (3-μm pores) following a 4 h exposure of HASMCs to an FBS gradient. Prior to exposure of these cells to the FBS gradient, the cells had been transfected either with a control siRNA (siCtrl) or an EPAC1-targeting siRNA (siEPAC) for 48 h. Actin (red) and nuclei (blue) were visualized by incubating fixed cells with TRITC-conjugated phalloidin or DAPI, respectively (scaling bars, 50 µm). Note: Since 3 µm pores precluded migration of HASMCs to the lower level of these transwells, no DAPI (blue) staining is present in these images. **(C)** Quantification of the LEPs formed by siCtrl or siEPAC1 transfected HASMCs following their exposure to the FBS gradient are shown. Statistically significant reduction in LEPs formed by siEPAC1 HASMCs compared to siCtrl HASMC was determined by comparing results obtained in n = 3 independent experiments using the Student’s unpaired *t*-test, **** *p* < 0.0001). **(D**–**E)** Quantification of the impact of inhibiting EPAC1 with CE3F4 (20 µM) **(D)** or activating EPAC1 with 8-CPT-2′-*O*-Me-cAMP (100µM) **(E)** on LEP formation in HASMCs in n = 3 independent experiments using the Student’s unpaired *t*-test, **** *p* < 0.0001.

**Figure 2 cells-08-01473-f002:**
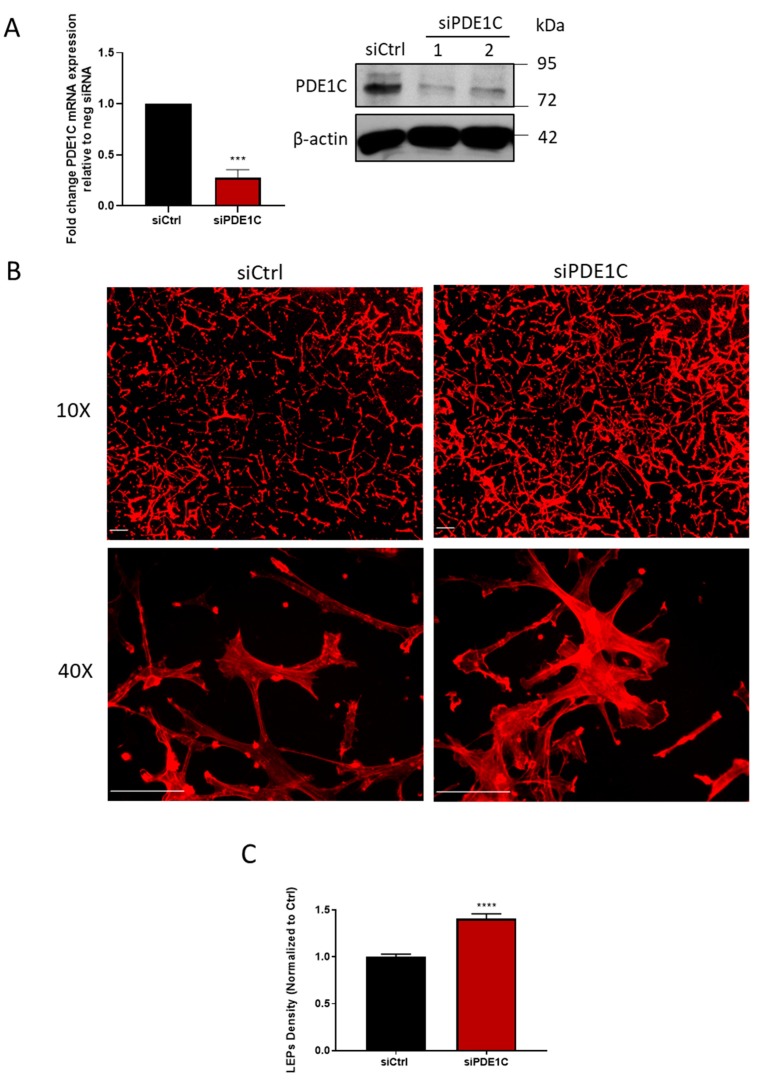
PDE1C silencing promotes HASMC LEPs formation. **(A)** Detection of PDE1C protein by immunoblotting (immunoblots), or mRNA by qRT-PCR (histogram), of samples obtained from a representative experiment (immunoblot) indicating the knockdown efficiency of PDE1C using 2 different siRNAs (PDE1C siRNA #1 or PDE1C siRNA #2) or n = 3 independent experiments (histogram), in which HASMC had been incubated with siCtrl or PDE1C siRNA #2 for 48 h. Reductions in PDE1C protein and mRNA were both statistically significant, as assessed using the Student’s unpaired *t*-test, **** *p* < 0.0001. **(B)** Representative images, obtained at either 10× or 40× magnification, of actin-stained LEPs detected on the lower levels (bottom) of FluoroBlok^TM^ transwell (3-μm pores) following a 4 h exposure of HASMCs to an FBS gradient. Prior to exposure of these cells to the FBS gradient, the cells had been transfected either with a control siRNA (siCtrl) or a PDE1C-targeting siRNA (siPDE1C) for 48 h. Actin (red) and nuclei (blue) were visualized by incubating fixed cells with TRITC-conjugated phalloidin or DAPI, respectively (scaling bars, 50 µm). Note: Since 3 µm pores precluded migration of HASMCs to the lower level of these transwells, no DAPI (blue) staining is present in these images. **(C)** Quantification of LEP formation in siCtrl or siPDE1C transfected HASMCs in response to exposure to an FBS gradient for 4 h in n = 3 independent experiments, assessed using the Student’s unpaired *t*-test, **** *p* < 0.0001.

**Figure 3 cells-08-01473-f003:**
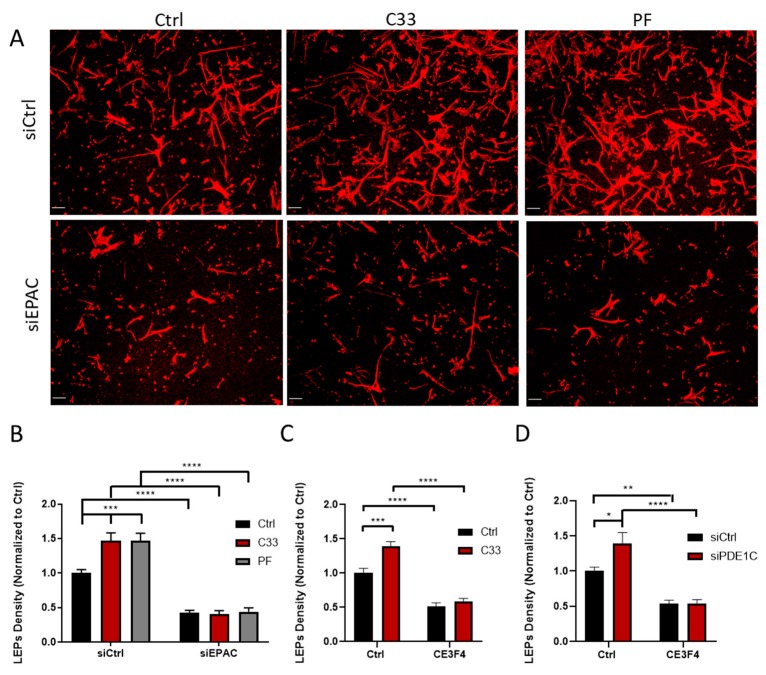
EPAC1 and PDE1C cooperate to allow HASMC LEP formation. **(A)** Representative images, obtained at 10x magnification, of actin-stained LEPs detected on the lower levels (bottom) of FluoroBlok^TM^ transwell (3-μm pores) following a 4 h exposure of HASMCs to an FBS gradient in the presence of vehicle (0.1% DMSO *v/v*) or this same concentration of DMSO containing C-33 (1 µM) or PF-04827736 (1 µM). Prior to exposure of these cells to the FBS gradient, the cells had been transfected either with a control siRNA (siCtrl), an EPAC1-targeting siRNA, or a PDE1C-targeting siRNA (siPDE1C) for 48 h. Actin (red) and nuclei (blue) were visualized by incubating fixed cells with TRITC-conjugated phalloidin or DAPI, respectively (scaling bars, 50 µm). Note: Since 3 µm pores precluded migration of HASMCs to the lower level of these transwells, no DAPI (blue) staining is present in these images. **(B)** Quantification in the density of LEPs in siCtrl or siEPAC1 transfected cells treated as above. Data from n = 3 independent experiments were normalized to appropriate controls and significance was calculated using a two-way ANOVA and the Tukey’s post-hoc analysis, *** *p* < 0.001, **** *p* < 0.0001. **(C)** Quantification of LEP density in HASMCs treated with DMSO (0.1% *v/v*) or this same concentration of DMSO containing C-33 (1 µM) in the presence or absence of CE3F4 (20 µM). Data from n = 3 experiments were normalized to the vehicle DMSO and significance was determined with a two-way ANOVA and the Tukey’s post-hoc analysis, *** *p* < 0.001, **** *p* < 0.0001. **(D)** Quantification of LEP density in siCtrl or siPDE1C HASMCs treated with DMSO (0.1% *v/v*) in the presence or absence of CE3F4 (20 µM). Data from n = 3 experiments were normalized to siCtrl control values and significance was assessed using a two-way ANOVA and the Tukey’s post-hoc analysis, * *p* < 0.05, ** *p* < 0.01, **** *p* < 0.0001. **(E**–**H)** The impact of transfecting HASMCs with either control siRNA, EPAC1, or PDE1C-targeting siRNAs, or incubating these cells with either PDE1 or EPAC1 inhibitors, either alone or together, on the density of HASMC on the upper levels of the FluoroBlok^TM^ transwells was measured. **(E)** Representative images, obtained at 10× magnification, of HASMCs on the upper levels (top) of FluoroBlok^TM^ transwells (3-μm pores) following a 4 h exposure of siCtrl or siEPAC1 transfected HASMCs to an FBS gradient in the presence of vehicle (0.1% DMSO *v/v*) or this same concentration of DMSO containing either C33 (1 µM) or PF-04827736 (1 µM). Staining for actin (red) and nuclei (blue) was carried out by incubating fixed cells with TRITC-conjugated phalloidin or DAPI, respectively (scaling bars, 50 µm). In each siCtrl and siEPAC1 transfected cell images, the top row shows both actin and nuclei while the lower row of images shows only the nuclei. **(F–H)** Quantification of the number of nuclei in a series of images obtained from transwells used in n = 3 independent experiments.

**Table 1 cells-08-01473-t001:** siRNA sequences.

Target	siRNA ID	Sense	Antisense
PDE1C # 1	PDE1CHSS107703	5′-UAUAGCAAAGAUCUCCAGCUCCGUC-3′	5′-CACCAGCUGUUA UUGAGGCAUUAAA-3′
PDE1C # 2	PDE1CHSS182019	5′-CACCAGCUGUUAUUGAGGCAUUAAA-3′	5′-UUUAAUGCCUCAAUAACAGCUGGUG-3′
EPAC1	RAPGEF3HSS115938	5′-CCUCAAGGAGCAGAAGAAUCUCAAU-3′	5′-AUUGAGAUUCUUCUGCUCCUUGAGG-3′

**Table 2 cells-08-01473-t002:** Primer Sequences.

Gene	Forward	Reverse
TBP	TATAATCCCAAGCGGTTTGC	GCTGGAAAACCCAACTTCTG
PGK	CTGTGGGGGTATTTGAATGG	CTTCCAGGAGCTCCAAACTG
PDE1C	CAGCAAAAGCATGGGACCTC	TGAAGGTGGGTTCCACGATG

**Table 3 cells-08-01473-t003:** LEP formation in HASMCs treated with PDE inhibitors.

Treatment	Density of LEPs on Bottom of Transwells (% of Control)
DMSO	100 ± 5
C33 1 µM	178 ± 12 ^1^
Ro 20-1724 10 µM	78 ± 14
Cilostamide 5 µM	46 ± 3 ^1^

Values are means of n = 3 independent experiments ± SEM. ^1^
*p* < 0.0001 compared with vehicle-treated HASMC LEP formation, as determined by a one-way analysis of variance (ANOVA) and Dunnett’s multiple comparisons test.

**Table 4 cells-08-01473-t004:** LEP formation in siRNA-treated HASMCs +/- the PDE1 inhibitor (C33).

	Density of LEPs on Bottom of Transwells (% of Control)	
	Treatment	
	DMSO	C33 1 µM
siCtrl	100 ± 7.1	175 ± 17 ^1,2^
si1C	196 ± 11 ^1,2^	192 ± 15 ^1,2^

Values are means of n = 3 independent experiments ± SEM. ^1^
*p* < 0.001 compared with vehicle-treated HASMC LEP formation, as determined by a two-way ANOVA and Tukey’s multiple comparisons test. ^2^
*p* > 0.05; means are not significant between siCtrl (C33) and si1C (DMSO) or si1C in the presence or absence of (C33).

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
