# Peer review of "An EPAC1/PDE1C-Signaling Axis Regulates Formation of Leading-Edge Protrusion in Polarized Human Arterial Vascular Smooth Muscle Cells"

_cells, 2019, doi:10.3390/cells8121473_

Round 1

Reviewer 1 Report

The authors present interesting data investigating the role of EPAC in leading edge protrusion and migration of human aortic VSMC. The data presented helps to address the contradictory data in the literature on the role of cAMP and EPAC in the  regulation of VSMC migration. They present convincing data indicating that EPAC1 and PDE1C are responsible for regulating leading edge protrusion formation in human aortic vascular smooth muscle cells. The data is generally convincing but several relatively minor points should be addressed by the authors:   1  The paper would benefit from a more detailed explanation of the LEP quantification method. How was the phalloidin fluorescence signal on the bottom of the membrane quantified? Presumably this was achieved by images analysis of the images as indicated in figure 1B? Does the total phalloidin signal in these images represent the number/density of LEP? In the representative images, there appears to be entire cells present? Is this the case and how does this effect the LEP quantification?   2 In figure 1C, D and E the LEP are quantified as LEP density. This implies LEP/unit area. Would relative LEP be more appropriate?     3 The degree of EPAC1 silencing (Figure 1A) is quite modest. I accept that HAVSMC are often difficult to transfect at high density but do the authors think this relatively small reduction in EPAC1 is sufficient to detect effects on LEP formation? Maybe the authors could comment on this?   3 The data presented in tables 3 and 4 may be easier for the reader to understand if presented as a graph rather than a table. Is there a reason why this data is presented in table form, while the LAP quantification in Fig is presented as a graph?    

In Figure 1E the authors show that activation of EPAC ith 8-CPT does not increase LEP in HAVSMC. The authors argue that this reflects the fact that EPAC is already activated in LEP and hence cannot be activated further. If this is the case, how does PDE1C silencing or pharmacological inhibition increase LEP formation? Presumably this is due to local increases in cAMP levels that activate EPAC1, as suggested by the data presented in Figure 3. Can the authors explain why this is? o the authors have any data to indicate that the 8-CPT treatment (Figure 1E) actually increased EPAC activity?

Author Response

Reviewer 1: We wish to thank the reviewer for his/her constructive critisms and highlight below our responses to these comments.

The Methods have been revised in order to more clearly delineate how the LEP were visualized and quantified.

àThe representative images are not whole cell images, rather they are leading edge protrusions (LEPs) as only the actin structures (stained in red) are observed and not the nuclei (stained in blue) on the bottom (underside) of the FluoroblokTM transwell membranes. The FluoroblokTM transwell membranes used in our study contain very small pores (3µm) which obviate movement of the cell nucleus onto the bottom of the transwell, thereby precluding whole cell migration.

Quantification in the formation of LEPs is made by measurement of the total density of all structures on the bottom of the transwell, not individual structures. As such, relative LEP per unit area was not measured, but rather total LEP density or actin fluorescence. This also is now more accurately described in the Methods.

We measured EPAC1 knockdown by immunoblot analysis, not mRNA, as this was a more accurate assessment of the reduced EPAC1 activity in our cells. Consistent with a 50% knockdown efficiency of EPAC1 protein, we observed roughly a 50% effect on inhibition of leading edge structure formation. We are of the opinion that this links the knockdown effectiveness to the formation of LEPs in our studies.

We chose to show these data in table form, rather than histograms, for purely stylistic reasons as all of the other data are in histograms. We are of the opinion that “switching things up”, would keep the readers interest. Indeed, we hypothesize that activation of EPAC1 by 8-CTP does not alter LEP formation in HASMCs because the relevant “pool” of EPAC1 is already activated during LEP formation; however, PDE1C silencing or inhibition increases LEP formation due to a local elevation in cAMP which activates this “pool” of EPAC1 in leading edge protrusions. Evidence suggests that the concentration of the EPAC activator 8-CPT (100µM) used is indeed activating EPAC as previously shown by our lab in the following paper: Cell Signal. 2016 June;28(6):606-19, where human aortic endothelial cells incubated with 8-CPT (100µM) were shown to significantly reduce the disruptive actions of fluid shear stress on adherens junctions.

Reviewer 2: We wish to thank the reviewer for his/her constructive critisms and highlight below our responses to these comments.

The concentrations of the PDE inhibitors used in the human arterial smooth muscle cells were based on previous studies in our laboratory and are the maximal concentrations that yield selective inhibition of their target as shown in several manuscripts, now referenced in the text (i.e. Circ Res. 1998 May 4;82(8):852-61 and Br J Pharmacol. 1997 Sep;122(2):233-40. PDE1 inhibitor (C33) concentrations were chosen based on data contained within the manuscripts (listed below) describing the synthesis and inhibitory profile of this compound as published by Dr Breitenbucher (Dart Neurosciences). Ahn, H.S., et al., Potent tetracyclic guanine inhibitors of PDE1 and PDE5 cyclic guanosine monophosphate phosphodiesterases with oral antihypertensive activity. J Med Chem, 1997. 40(14): p. 2196-210 Dyck, B., et al., Discovery of Selective Phosphodiesterase 1 Inhibitors with Memory Enhancing Properties. J Med Chem, 2017. 60(8): p. 3472-3483.

Similarly, PF-04827736 concentrations were chosen based on the following (Humphrey JM, et al. J Med Chem. 2018 May 10. doi: 10.1021/acs.jmedchem.8b00374)

The concentration of the EPAC inhibitor CE3F4 (20µM) and of the EPAC activator used in our study was based on previous work in our laboratory (Cell Signal. 2016 June;28(6):606-19; Sci Rep. 2019 Feb 20;9(1):2385. doi: 10.1038/s41598-018-37805-y) and work of others including the laboratory of Dr. Fischmeister. We agree and have used 2 PDE1 inhibitors and each were shown to mediate effects on LEP shown in Fig. 3 using compounds (C33) and (PF-04827736) We measured EPAC1 and PDE1C knockdown at 48 h post siRNA addition as stated in the Methods. We used immunoblot analysis (EPAC1), and both protein and mRNA for PDE1C, as protein was thought to be a more accurate assessment of the reduced protein activity in our cells. Indeed, the level of knockdown of EPAC1 was ~50% which is consistent with the findings that there is ~50% reduction of LEP formation. PDE1C and PDE1A, but not PDE1B, are expressed in vascular smooth muscle cells from several species, including humans, and this is stated in the manuscript (Curr Opin Pharmacol. 2011 Dec;11(6):720-4. doi: 10.1016/j.coph.2011.09.002. Epub 2011 Sep 29) and (J Clin Invest. 1997 Nov 15;100(10):2611-21.). Notwithstanding the expression of PDE1A, we focused our attention on PDE1C for this work because PDE1C is known to hydrolyze cAMP much more efficiently than PDE1A. Indeed, the Km for cAMP for PDE1A and PDE1B genes is ~113µM for PDE1A and ~24µM for PDE1B, while the Km for cAMP for PDE1C is ~1-3µM as stated (Calmodulin and Signal Transduction. 1998, Pages 237-286. https://doi.org/10.1016/B978-0-08-092636-0.50009-8). We have previously determined in our lab that silencing PDE1C does not alter PDE3/PDE4 activity (alternate manuscript in review). However, we have not investigated the impact of silencing PDE1C on the levels or PDE1A expression and this will require future investigations. Silencing of PDE1C in vascular smooth muscle cells has been shown in the literature to disrupt cell motility as shown (Circ Res. 2015 Mar 27;116(7):1120-32. doi: 10.1161/CIRCRESAHA.116.304408. Epub 2015 Jan 21) as well as detected in our lab (alternate manuscript in review). It has also been shown that silencing PDE1C inhibits migration in glioblastoma multiforme cells as shown (Mol Carcinog. 2016 Mar;55(3):268-79. doi: 10.1002/mc.22276. Epub 2015 Jan 25.). We have previously determined in our lab that silencing of PDE1C expression significantly reduces Ca2+/CaM stimulated cAMP PDE activity compared to control siRNA transfected human arterial SMCs (alternate manuscript in review). The selective staining of the very low abundant endogenous levels of EPAC1 or PDE1C is simply not possible given the limited avidity of the available antisera. Discussion: We have further added to our discussion to include the potential calcium regulation of leading-edge protrusion via the PDE1C/Epac1 axis.

Reviewer 3: We wish to thank the reviewer for his/her constructive critisms and highlight below our responses to these comments.

The culture of human arterial smooth muscle cells is cited in the methods: Ochsner J. 2007 Fall;7(3):133-6. (Isolation of endothelial cells and vascular smooth muscle cells from internal mammary artery tissue.) PDE1 inhibitor (C33) concentrations were chosen based on data contained within the manuscripts (listed below) describing the synthesis and inhibitory profile of this compound as published by Dr Breitenbucher (Dart Neurosciences). Ahn, H.S., et al., Potent tetracyclic guanine inhibitors of PDE1 and PDE5 cyclic guanosine monophosphate phosphodiesterases with oral antihypertensive activity. J Med Chem, 1997. 40(14): p. 2196-210; Dyck, B., et al., Discovery of Selective Phosphodiesterase 1 Inhibitors with Memory Enhancing Properties. J Med Chem, 2017. 60(8): p. 3472-3483.The C33 compound was derived from the synthesis of the following papers: Mea culpa, we did inaccurately describe our statistical methods. Indeed Table 3 and Table 4 contained typos, which have been corrected.

The discussion has been elaborated upon to include the relevant clinical significance of our findings as well as the importance in the study of PDE1C in human arterial smooth muscle cells.

Reviewer 2 Report

            This is an exciting, highly detailed, extremely thorough and well-performed study that uncovers a novel regulatory node involving the Epac cAMP detector system and the calcium-activated, cAMP-degrading phosphodiesterase-1C enzyme. This is shown to have relevance to a fundamental cellular process, namely in regulating leading edge protrusion (LEP). As such, it can be expected to be of wide and general interest.

            Cells express a panoply of cyclic nucleotide phosphodiesterase (PDE) enzymes able to undertake the same transformation: namely to degrade cAMP.  However, such enzymes are able to control specific cellular processes by virtue of being targeted to specific signalling complexes. Thus, the relative abundance of specific PDE species in cells does not provide an indication as to their (relative) importance. Even a low abundance PDE can exhibit critical functionality when targeted to a specific intracellular site, as there it can regulate local cAMP levels, establishing gradients that control the activity of any similarly targeted cAMP effectors, such a Epac and PKA.  

            This study not only provides the first demonstration of the functional targeting of PDE1 but also the first demonstration that a PDE1 isoform regulates a cellular process through Epac1.

Minor comments.

Please provide the justification for use and the concentrations employed plus associated explanation, with references, for the use of selective inhibitors of the various PDE species and Epac. While the PDE3 and PDE4 inhibitors employed in this study are well-recognised, there can be some ambiguity over the selectivity and potency of pDE1 inhibitors. Could the authors please discuss this, especially in the light of PDE1 isoforms, particularly PDE1C, and consider using more than one type of PDE1 inhibitor.

Please expand on the silencing approaches used: primers, effect on mRNA levels and time-courses etc.

(a) The level of KO of Epac1 (Fig 1A) was around 50%. Do the authors believe that this really sufficient to affect functionality per se? Presunably if the effect of 50% knockdown matches the phenotypic effect of 50% pharmacological inhibition then the authors woudl feel re-assured?  

How does this relate to levels of mRNA for Epac1?

Is the turnover of Epac1 low and so would a longer time-course of action aid in KO?

(b) In contrast to the Epac1 silencing, the PDE1C silencing was dramatic.  

Is PDE1C the only form of PDE1 in these cells? 

Does KO of PDE1C cause up-regulation/induction of PDE1A/PDE1C or PDE3/PDE4 isoforms in response to this disruption?

Does PDE1C silencing cause any change in PDE activity (total, calcium-stimulated, PDE1 inhibitor-inhibited levels?)

Does PDE1C knockdown alter cell mobility / polarity?

Is there either co-localisation or co-immunoprecipitation of PDE1C and Epac1

Discussion - Would the authors like to comment upon whether the PDE1C/Epac1 axis might provide a node for calcium regulation of leading-edge protrusion? There is an interesting article (Science Reports 2018 Feb 5;8(1):2433) showing that an intracellular calcium signal at the leading edge regulates mesodermal sheet migration during Xenopus gastrulation play an essential role in the active cell migration during gastrulation, for example. Also that the graded distribution of intracellular second messengers, such as Ca(2+) and cAMP, mediates directional cell migration, including axon navigational responses to extracellular guidance cues, in the developing nervous system (J Neuroscience 2016 May 18;36(20):5636-49). Such a system as the authors have uncovered here may also have implications for metastasis (Biomed Res Inter 2015;2015:409245).

Author Response

(The authors gave the same response as above.)

Reviewer 3 Report

The subject of the article is very interesting, once a more global effects of EPAC1 in ASMCs is explored. This article show that inhibiting EPAC1 activity, or markedly reducing its expression, inhibited LEP formation by migrating HASMCs.

General comments:

In the Materials and Methods, I didn't understand the cell culture? Pleased further explain or add a bibliographic reference for this culture. The Compound 33 (C33) is a specify inhibitor for PDE1C? what is the chemical formula? was synthesized by Dr. Guy Breitenbucher? In the table 3 why the authors used one-way ANOVA and Tukey’s multiple comparisons? Once DMSO is the control the Dunnett's test is better. In the table 4, the statistic is between the DMSO and C33 1μM, why one-way ANOVA? There are only two parameters? In my opinion it is necessary to discuss the clinical importance, which could give even a greater relevance to this work. The Discussion need to be improved, regarding the PDE1C? Why this PDE1, is the unice in this cells?

Author Response

(The authors gave the same response as above.)

Round 2

Reviewer 3 Report

The authors have revised the manuscript according to the reviewer's comments.

I recommend accept